# Telemedicine as an Untapped Opportunity for Parkinson’s Nurses Training in Personalized Care Approaches

**DOI:** 10.3390/jpm12071057

**Published:** 2022-06-28

**Authors:** Marlena van Munster, Johanne Stümpel, Timo Clemens, Katarzyna Czabanowska, David J. Pedrosa, Tiago A. Mestre

**Affiliations:** 1Department of Neurology, University Hospital Marburg, 35043 Marburg, Germany; david.pedrosa@uni-marburg.de; 2Department of International Health, CAPHRI Care and Public Health Research Institute, Maastricht University, 6229 ER Maastricht, The Netherlands; timo.clemens@maastrichtuniversity.nl (T.C.); kasia.czabanowska@maastrichtuniversity.nl (K.C.); 3Cologne Center for Ethics, Rights, Economics, and Social Sciences of Health (CERES), University of Cologne, 50931 Cologne, Germany; johanne.stuempel@uk-koeln.de; 4Center for Life Ethics, University of Bonn, 53113 Bonn, Germany; 5Department of Health Policy Management, Institute of Public Health, Faculty of Health Sciences, Jagiellonian University, 31-066 Krakow, Poland; 6Parkinson Disease and Movement Disorders Centre, Division of Neurology, Department of Medicine, The Ottawa Hospital Research Institute, University of Ottawa Brain and Mind Research Institute, Ottawa, ON K1Y 4E9, Canada; tmestre@toh.ca

**Keywords:** Parkinson’s disease, nursing training, telenursing, telemedicine, Parkinson nurse, personalized care, competency development

## Abstract

(1) Background: Parkinson nurses (PN) take over important functions in the telemedical care of person’s with Parkinson’s disease (PwPs). This requires special competencies that have so far been largely unexplored. The aim of the article is to identify competencies of PN operating in a personalized care model. (2) Methods: This study employed a qualitative approach. Based on the competency framework for telenursing, PN were asked about their competencies using a qualitative online survey. (3) Results: The results show that PN need competencies on a personal and organizational level, as well as in the relationship with PwPs. PN have developed these skills through professional exchange, training, and personal experience. In addition, both hindering and beneficial factors for the development of competencies could be identified. (4) Conclusions: Competency development for telemedical care is complex and must be designed and promoted in a targeted manner.

## 1. Introduction

Parkinson’s disease (PD) is a complex non-curable neurodegenerative disorder presenting motor and non-motor features such as cognitive impairment, mood and sleep disorders, autonomic dysfunction, and pain [1]. This clinical complexity implies that individualized care delivery is needed [2]. Nevertheless, to date, personalized care delivery models for persons with PD (PwPs) are rare [3]. A key aspect for delivering personalized care is the availability of specialized staff [3,4]. In addition to medical expertise, the implementation of personalized care structures also requires skills in the areas of networking, coordination and patient centricity [2,5,6,7]. Since these skills may not be part of the classical training of healthcare professional, the question arises as to which profession could take on the role of the facilitator. It has been recognized that Parkinson nurses (PN) may play this key role [5,8]. PN can accomplish important tasks in the care process, which are not taken over by any other profession, such as patient navigation [5,7]. In practice, there is no uniform understanding of the PN [8,9] or a firmly defined framework of competencies, even though specialized training for PN on the delivery of personalized care services has been recommended [7]. 

The “Framework for Training Parkinson Nurses” to deliver a personalized care approach addresses this gap [7]. The framework was created based on requirements, practical models, and theoretical concepts of personalized Parkinson’s care and thus includes various perspectives on sustainable Parkinson’s care.

Concerning the fourth module, the framework is deliberately kept open, since the application of telemedicine takes place in many ways, and thus requires context-dependent competencies. However, it appears desirable to have examples of how competencies for this module could be defined since more and more telemedical solutions for PD are on the rise. As in other areas of PN training,, clear competency profiles for telenursing have been lacking. It is not only through the COVID-19 pandemic that virtual patient care trends seem to be rising [10]. Telemedical solutions, if implemented appropriately, may offer many benefits to PwPs and care providers, such as faster communication and closer monitoring possibilities [11]. However, recent studies also show that telemedicine solutions are not a sure-fire success, but require a custom-fit, easy-to-understand and well-supervised implementation, and thus, special expertise [12]. Returning to the idea that PN take over important roles in the care process that facilitate personalized Parkinson’s care, they may be important enablers of virtual care [13,14,15]. However, there has been little insight into which competencies nurses who work with a telemedicine need, in order to enable virtual patient care. The article aims to address this knowledge gap by evaluating the skills and expertise of PNs working in a technology-supported, home-based, community-centered, integrated care model (iCARE-PD) [16].

## 2. Materials and Methods

### 2.1. Study Design and Theoretical Background

This study employed a qualitative approach to explore the perspectives of PN about the competencies that are required for utilizing telemedicine solutions. A qualitative approach was considered appropriate for this pilot study as it enables in-depth insights into the professional skills, perspectives and expertise of PNs in regard to telehealth. The greatest challenge was to design the study in such a way that it was possible for the PN to answer the questions in a time-efficient manner. In addition, there was the difficulty that competencies are a construct that is difficult to grasp, which is why it was considered necessary to introduce the participants to the concept. Compared to classic qualitative survey methods, online surveys offer the advantage that they enable a diverse presentation of input, and the respondents have more time to reflect on their answers, since they are not under pressure to formulate an answer quickly or to fill in pauses [17,18]. For the given context and the formulated challenges, we adopted a qualitative online survey [17].

In order to make the construct of competency more tangible, we based the survey on van Houwlingen et al.’s competencies for telenursing [19]. According to the authors, competencies are reflected in three central domains: skills, knowledge and attitude [19]. Through a Delphi consensus approach, the authors have formulated 52 competencies that nurses need to have to deliver telemedical supported health care. The current study was therefore designed as follows: first, the competencies of a domain (skill, knowledge, attitude) were presented to the PN. Next, the PN were asked to evaluate the statements in terms of relevance to their work. Third, the nurses were instructed to select the three most important competencies for each domain. These steps should serve to enable an in-depth exploration of the idea of a competency. Subsequently, the nurses were then asked open-ended questions for each domain. A structure of the survey is shown in Table 1.

### 2.2. Data Collection

Data collection took place in the first quarter of 2022. The invitation to the survey was sent via an e-mail. The survey was accessible via a link, which the respondents could use to answer the survey anonymously. It was also possible to pause the survey and continue it at a later point in time.

### 2.3. Data Analysis

Data from the open-ended questions were analyzed by two team members (M.vM. and J.S.) using thematic analysis, as described by Braun and Clarke, which includes the following steps [20]:Data Familiarization: All responses were read repeatedly and discussed by the research team members to obtain a first impression of the data.Initial Code Generation: Relevant text units were extracted and given a label.Search for Themes: All codes were divided in broad, overarching themes based on similarities.Theme Review: The final themes were organized in a coherent pattern with a focus on internal and external homogeneity.Theme Definition: A brief description for each theme was created.Report Production: The final report was created.

To ensure the credibility of the results, the initial code generation and theme search was performed independently by the research team members. Discrepancies were resolved through mutual discussion. All further steps were completed in joint discussion rounds. The qualitative data presented here has been conducted in accordance with the “Consolidated criteria for reporting qualitative research (COREQ)” [21].

## 3. Results

In total, six surveys out of nine invitations sent were filled out completely and form the basis for the results. Participants were nurses, providing telemedicine care to PwP as part from the iCARE-PD study in Canada and Europe (see www.icarepd.ca (accessed on 1 July 2021). The participants had between < 1 and 5–7 years of professional experience as a PN, and worked between 25% and 100% of their job in this role. Other sample characteristics can be retrieved from Table 2. 

In the knowledge dimension, the nurses rated understanding technology application, and, on the other hand, understanding of the context in which telenursing is provided, as particular important (cf. Table 3).

Within this dimension, knowledge regarding technology application and knowledge of relevant laws or regulation were mentioned most frequently by the nurses. As one participant summarized:


*“(…) knowing the clinical process is essential because this will allow one to determine which tools in telemedicine could be helpful in providing care to the patients. Particularly in movement disorders and PD (Parkinson’s disease) there are certain limitations that one must consider such as age of the participants, degree of disability, and access to technologies. Keeping this in mind is important otherwise using certain telemedicine tools might not be appropriate. Lastly, with the strict implementation of GDPR (General Data Protection Regulation) (…)—it is essential to know what data protection rules are involved with introducing/using new technologies and engaging in telemedicine.”*


In the attitude dimension, the nurses identified an empowering attitude towards PwPs as being highly relevant (cf. Table 4). 

One participant described the connection between personal attitude and attitude towards PwPs as follows:


*“I think, first of all, it is necessary that I myself am open to the use of digital technologies and have confidence in these techniques or that their use brings benefits for everyone (…). Because then I can act much more authentically and motivated.”*



*“I feel that the Parkinson’s population (…) [is] not as comfortable with technology. It is important to encourage and teach them with positive feedback that being seen virtually does not take anything away from the appointment and that they still benefit and get good helpful information. I encourage to see the positives that they can stay in the comfort of their own home making it easier on them and their caregiver by decreasing the need for tiring transportation.”*


In the third and last competency dimension, “skill”, the nurses named the ability to recognize patient needs from a distance (*n* = 3) as an important skill. For this purpose, the existence of basic technical expertise (*n* = 2) and the ability to critically analyze (*n* = 1) were rated as important skills. In addition, the ability to incorporate clinical knowledge into decisions (*n* = 2) was rated an important skill. The ability to instruct the patients in the use of technologies (*n* = 1), the ability to create a confidential environment online (*n* = 1), the ability to observe well (*n* = 2), to listen carefully (*n* = 2), to protect patient privacy (*n* = 1), and to be able to deal with patient concerns regarding the use technology (*n* = 1) were mentioned as further important skills. Based on this, the ability to decide whether a technology is comfortable for a patient (*n* = 2) was also considered a particularly important skill. One participant described the connection between observation and application skills as follows:


*“When using digital technologies, I must have the ability to ensure security and trust which means I have to ensure that the privacy of the participants is protected and that I can convey. It is also very important for me to make a realistic assessment of which technology is used based on the participant’s preferences, so that the participant feels taken seriously and stays motivated. This requires good observer-listening qualities.”*



*“I feel that with virtual visits you need to be acutely aware, listen carefully and watch for visual cues when assessing the patient’s related concerns.”*


The nurses used various resources for the acquisition of competencies. The most frequently mentioned resources included a curriculum that had been specially completed for the care concept [7]. In addition, the experience from the nurses’ own work and the exchange with colleagues was an important resource: 


*“In general, since this figure does not exist in [my country], the most important thing for me was the specific training (see [7]) and the comparison with international colleagues.”*



*“I acquired the knowledge through shadowing a PD nurse specialist and attending Movement disorder clinics. I’ve also benefited from the ICare-PD nurse curriculum (see [7]) and the various meetings with [colleagues] to discuss the topics covered in the curriculum.”*



*“A lot of self-learning to understand this disease as well as the practical experience obtained working as a movement disorder nurse. I have (and still) read and study the disease to ensure that I am up to date.”*


The iCARE-PD intervention also includes the provision of education materials for PwPs and care partners. Studying its content was also described as a valuable resource. Other resources used for competency development were online resources, such as video tutorials, the use of personal experience, family support and training offers from the employer. 

In addition to the resources, the nurses also named some factors that can be beneficial or that can hinder the development of competencies. As facilitators, the nurses indicated sufficient time to deal with the technologies, appropriate training, and information materials:


*“I did not have many difficulties (to acquire relevant knowledge) because I had the right time and good material (..)”*



*“It is about having the time.”*


The COVID-19 pandemic and the need to continue to provide care were described as push-factors, making it essential to deal with various digital technologies that are used in patient care:


*“Corona” has certainly triggered certain aspects of telemedicine and meant that I had to deal with virtual communication formats (zoom conferences, joint use of platforms for sharing and editing information, etc.).”*



*“Since competent handling of (…) technologies was/is essential for [my] work—this factor has a “motivating” effect.”*


The partial incomprehensibility of software and instructions in languages other than the native language was described as an obstacle to the acquisition of technological competency:


*“In connection with the acquisition of knowledge in dealing with (a) database, some language problems and the resulting misunderstandings have arisen. It sometimes took several emails (hence patience) until questions could be clarified. Using [video conferencing system] initially proved difficult.”*


Finally, the nurses described a few factors that could be both beneficial and hindering in the development of competencies (see Figure 1).

Depending on the personal attitude, the acquisition of skills could be perceived as more difficult or easier:


*“The sum of my own attitudes and believes is made up of different aspects. This includes your own personality, openness and willingness to learn (…)”*



*“Adaptation to new technologies specifically requires both parties (patients and PD nurses) to invest a significant time and effort. Moreover, one must be open to innovation and change because if the PD nurse doesn’t carry this mindset, it would be hard to push initiatives for change.”*


The environment in which the technology was dealt with could both promote and hinder the development of competencies and influence personal attitude:


*“One of the struggles in acquiring these (right) attitudes is if you work in an environment that does not value change and innovation. It can be quite hard to have a positive attitude towards digital technologies if the place where you work does not care for these things. For me, I choose to leave such a work place and find employment where they value change and innovation.”*


In addition to this general environment, the opportunity for the practical application of telemedicine was an important factor. Nurses who use a new technology often and a lot found it easier to develop new skills:


*“One of the struggles I faced was in regards to finding practical application that is specific to the cohort of patients that we see at our site.”*



*“The thing that limits me most, however, is not being able to put this knowledge into practice very much, precisely because my job role at the moment is associated almost exclusively with clinical trials, but is not applied to real clinical practice.”*



*“The acquisition of (my) skills is based on professional and life experience as well as learning how to use digital technologies and, above all, regularly applying what I have learned.”*


The last important factor was the start of using technology in life. While developing telemedicine skills was easy for participants born in the digital age, participants who started using technology later in life described certain connections that were more difficult to understand:


*“The handling and use of digital technologies only played a role late in my professional and private life. So, it was not so easy for me to learn how to use it, because it was breaking new ground and differed from previous knowledge and experience.”*



*“I think in general proficiency and digital literacy is an important skill to have when dealing with digital technologies as a PD nurse. It will be extremely difficult for anyone to navigate the digital space if they have poor digital literacy. (…) I think being born during the age of the internet is one the biggest factors that have allowed me to gain the proper skills that is needed to navigate this digital space (…)”*


## 4. Discussion

The present study has shown that the use of telemedical applications for Parkinson’s care requires specific competencies which go beyond just “knowing” what to do. Virtual Parkinson’s care is in fact a very heterogeneous field, so competencies must be developed to be context- and technology-specific. However, the present study also suggests that, in addition to targeted training, a supportive environment may strengthen the development of competencies as well.

For the context of the iCARE-PD project considered here, it could be shown that a subdivision of competencies into knowledge, skills and attitude enables an in-depth description and analysis of them. In all dimensions, it requires abilities that are closely oriented to the needs of PwPs. Although the use of telemedical applications in Parkinson’s care is not only carried out by PN, it seems crucial that a special patient proximity is created during the application. While patient centeredness is often the basic idea behind many telemedicine solutions [22], its success is not guaranteed [23]. Patient centricity is created, on the one hand, by technology that is easy to use for PwPs [24], and, on the other hand, by staff that accompanies PwPs in the process of telemedicine healthcare [12]. Telemedicine solutions are often part of complex care models that require staff to navigate PwPs through this complexity [12]. Telemedical solutions can hold numerous potentials for PwPs, but they need to be used in a meaningful way. In the future, the question will not only arise as to how care providers can be trained for this purpose [25,26], but also which profession can take on the role navigating PwPs through the care process [7]. In addition, it seems important to investigate usability criteria of digital technologies for PD care to support PwPs more in their self-management.

The role of the PN has not yet been clearly defined [7], but it has great potential in the field of telemedical-supported care [10,12]. For a better definition and consolidation of this role, clear descriptions of competencies are required. The results of the present study show that PN not only have PwPs in mind, but also clinical processes and larger contexts. However, how can the development of skills for nurses be promoted?

For the nurses, the central pillar for competency development in this study was specialized training and exchange with colleagues, as well as hands-on training. Consequently, they named the opportunity to apply what they had learned and to ask questions as the most important supporting factors in their competency development. In other areas of nursing professionalization, it was also found that the possibility of regularly applying what has been learned is a central factor for learning success [25,27]. Another factor that was mentioned as an important facilitator of competency development by the nurses was the motivation to learn and the willingness to try out new things. Studies from other areas of nursing training have shown that technology acceptance is a key component of learning success [28,29]. Therefore, it seems important not only to give the nurses the opportunity to use a certain technology, but also to explain its functions in a meaningful way and motivate them to use it.

For the future of PN training, it may therefore be desirable not only to define competencies for virtual and non-virtual Parkinson’s care, but also to create structures in which the nurses can regularly apply them and receive feedback.

### Strength and Limitations

This qualitative research was methodically designed and implemented to maximize the trustworthiness of the results. We have gained an understanding of the competencies that PN need in a personalized, telemedicine-supported care model, and how they developed these competencies. The limitations of this study included a limited context (an iCARE-PD study) and, as a result, a rather small sample size. The participants declared from 1 to 3 years of experience in the field of Parkinson nursing, which limits their professional experience. As the survey applied a non-disease specific framework for telenursing, no PD specific telemedicine concept was analyzed. Another limitation was that not all participants completed the survey, and therefore three data sets could not be included in the analysis. Further research may include PN from other personalized PD care models.

## 5. Conclusions

This study shed light on the field of competencies PNs may need in a technology-supported personalized care approach, and their perception of what helped them to develop these competencies. In this study, a personal attitude, a supportive environment, and the opportunity for practical application appear to be central promoters in the acquisition of skills. Applying technologies continues to take up more and more attention in the care of PwPs; therefore, PNs should be adequately prepared. In the future, there may need to be a clearer definition of the role of the PNs in telemedicine care to be able to better describe competencies and provide adequate training.

## Figures and Tables

**Figure 1 jpm-12-01057-f001:**
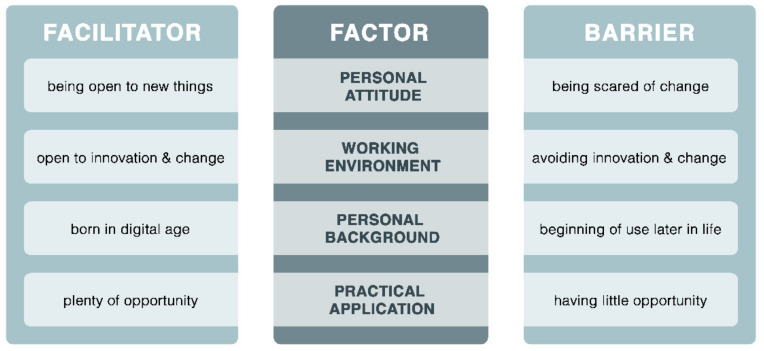
Factors influencing the development of competencies in the field of telenursing.

**Table 1 jpm-12-01057-t001:** Structure of the survey.

Section	Description	Content/Question
Professional Background	n.a. ^1^	highest level of education nursing training received official job title working experience as PN ^2^ in years percentage of job responsibilities as PN ^2^
Knowledge	Presentation of competencies according to [19]	Rating each competency on a Likert scale from 1 = little relevance to 5 = very high relevance.
		Choosing the 3 most relevant competencies:
	Open-ended questions	When you think of all statements regarding KNOWLEDGE—is there a certain type of knowledge that you consider or imagine as particularly important when dealing with telemedicine as a Parkinson nurse? If you now think about the knowledge that you need as an iCare-PD nurse: Can you please tell us how you acquired this KNOWLEDGE? (i.e., your formal education, personal experience...) What did you help the most to acquire this KNOWLEDGE? (i.e., reading books, practical examples, supervision...) 3. Can you please tell us if you faced any struggles/problems or obstacles when trying to acquire the relevant KNOWLEDGE? If so: Can you please tell us which struggles/problems or obstacles you faced and what you do to overcome these problems/struggles? How did you manage to acquire the knowledge?
Attitudes	Presentation of competencies according to [19]	Rating each competency on a Likert scale from 1 = little relevance to 5 = very high relevance
		Choosing the 3 most relevant competencies:
	Open-ended questions	When you think of all statements regarding ATTITUDES—is there a certain type of knowledge that you consider or imagine as particularly important when dealing with telemedicine as a Parkinson nurse? If you now think about the knowledge that you need as an iCare-PD nurse: Can you please tell us how you acquired these ATTITUDES? (i.e., your formal education, personal experience...) What did you help the most to acquire these ATTITUDES? (i.e., reading books, practical examples, supervision...) 3. Can you please tell us if you faced any struggles/problems or obstacles when trying to acquire the relevant ATTITUDES? If so: Can you please tell us which struggles/problems or obstacles you faced and what you do to overcome these problems/struggles? How did you manage to acquire these attitudes?
Skills	Presentation of competencies according to [1]	Rating each competency on a Likert scale from 1 = little relevance to 5 = very high relevance
		Choosing the 3 most relevant competencies:
	Open-ended questions	When you think of all statements regarding SKILLS—is there a certain type of knowledge that you consider or imagine as particularly important when dealing with telemedicine as a Parkinson nurse? If you now think about the knowledge that you need as an iCare-PD nurse: Can you please tell us how you acquired these SKILLS? (i.e., your formal education, personal experience...) What did you help the most to acquire these SKILLS? (i.e., reading books, practical examples, supervision...) 3. Can you please tell us if you faced any struggles/problems or obstacles when trying to acquire the relevant SKILLS? If so: Can you please tell us which struggles/problems or obstacles you faced and what you do to overcome these problems/struggles? How did you manage to acquire these skills?

^1^ n.a. = not applicable, ^2^ PN = Parkinson nurse.

**Table 2 jpm-12-01057-t002:** Sample description.

Sample Characteristics	
Professional training	Specialized nursing training (*n* = 2) University degree (*n* = 3) General vocational qualification (*n* = 1)
Current job position	Clinical nurse (*n* = 3) Study nurse (*n* = 3)
Experience with Parkinson’s disease	<1 year (*n* = 2) 1–3 years (*n* = 3) >5 years (*n* = 1)
Share of total working hours working as Parkinson nurse	up to 50% (*n* = 4) more than 50% (*n* = 2)

**Table 3 jpm-12-01057-t003:** Most relevant knowledge for telenursing as chosen by the Parkinson nurses.

Knowledge Relevant for Technology Application	Knowledge Regarding Context of Technology Application
technology application including clinical limitations (*n* = 4) patient’s preferences regarding technology (*n* = 2) technology implementation in one’s own institution (*n* = 1) technology application for collecting patient data (*n* = 1) relevant protocols in one’s own facility (*n* = 2)	applicable laws and regulations (*n* = 4) clinical processes within the institution (*n* = 2) knowledge of which health information online can be considered as reliable (*n* = 1)

**Table 4 jpm-12-01057-t004:** Most relevant attitudes for telenursing as chosen by the Parkinson nurses.

Attitudes Relevant for Technology Application
empowering patient to use telemedicine (*n* = 6) being honest, sincere and professional with patients (*n* = 3) being patient (*n* = 2) conveying empathy towards patients via digital channels (*n* = 2) having a self-confident and open attitude towards new technologies (*n* = 2) evaluating digital patient data in a standardized way (*n* = 2) motivating patient to use technology (*n* = 1)

## Data Availability

Data supporting reported results can be found in the following repository: doi: 10.5281/zenodo.6531503.

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
