# Peer review of "Telemedicine as an Untapped Opportunity for Parkinson’s Nurses Training in Personalized Care Approaches"

_jpm, 2022, doi:10.3390/jpm12071057_

Round 1
Reviewer 1 Report
Dear Authors,
The work is relevant and timely. However, it is suggested to review some points.
The introduction lacks sufficient references to previous work by other authors. In doing so, the authors fail to describe current limitations and the need for their contributions to the field.
The conclusion is quite positive regarding the results, considering the small number of study participants. It mentions: “This study showed what competences RNs need in a personalized care approach in the application of telemedicine and how they developed these competences”. It is suggested that the authors consider whether their contributions shed light on the field or definitively resolve the problem at hand.
Finally, the text should be reviewed. Some suggestions that would improve readability (this list is not exhaustive):
- Line 52. Start a new paragraph beginning with "The 52 Framework for Training...". Then, on line 57, instead of "Concerning the fourth module...", use "Concerning its fourth module...".
- Line 58. many different = many (or different).
- Line 62. I would suggest removing the expression “so far”. The sentence beginning with “It is not…” can be improved for clarity. I would suggest “… COVID-19 pandemic that virtual patient care trends seem to be rising [6]”.
- Line 69. Instead of telemedical-supported care, technology-supported care, telemedicine care, or remote care.
- Line 70. Not “the their”, just “the”.
- Line 76. I would suggest substituting “telemedical” with “telemedicine”.
- Line 81. A comma is missing.
- Line 98. There should not be a break between the table and the caption of that same table. Also, in tables, I suggest using only one font size.
- Line 101. Instead of “The data collection…”, “Data collection”.
- Line 106. The sentence is not clear. I would suggest “Data from the open-ended questions were analyzed by two team members (M.vM. and J.S.) using thematic analysis, as described by Braun and Clarke, which includes the following steps [13]:”.
- Line 122. The end of the sentence is missing.
- Line 127. Instead of “In the dimension of…”, “In the knowledge dimension…”. The rest of the paragraph also needs revision. For instance: This sentence is meaningless or obvious: “In the dimension of knowledge, the nurses rated knowledge as particularly relevant…”.
- Line 130. All rows in the table start with “knowledge…”. This can be improved. For instance, instead of “knowledge regarding technology application including clinical limitations”, “application of technology including clinical limitations”.
- Line 131. …as well as (no comma) the (missing “the”) in-depth…
- Line 137. “…as age or participants…” or “age of the participants”?
- The paragraph starting at line 142 is confusing. I suggest using a table to describe the results.
- Line 162. …as an important skill.
- Line 168. …considered an important skill.
- Line 172. …and that I can convey that.
- Line 177. …patient’s…
- Starting at line 182, the article transcribes the response of a study participant. Does the participant mention a reference as implicit in “(see [5])”?
- Line 190. The authors mention educational resources apparently developed by members of the iCARE-PD intervention. Are these resources available? Where?
- Line 126. The sentence is not clear. Please consider the following: “… some factors that can be beneficial or that can hinder the development of competences…”.
- Line 233. The sentence beginning with “The general…” is unclear. Moreover, . Please, consider the following: “In addition to this general environment, the opportunity for the practical application of telemedicine was an important factor.
- Figure 1. Different font sizes.
- Line 222. Repeated word.
- Line 234. …who use new technology…
- Lines 309 and 310. Please, consider “competence” and “competencies” as alternatives “competences”.
Author Response
Dear Expert,
Thank you for dedicating your time and expertise to reviewing our manuscript and providing such an extensive report.
We have revised the introduction and added additional references. Also, we have pointed out the knowledge gap better. We agree that the sample size is rather small, however, we still think that the provision of practice insights is adding a valuable contribution to the field. One reviewer recommended adding a sample description to the paper, which is why we have added information about the sample in Table 2.
In the care model that we have investigated 9 Parkinson Nurses provide telemedicine care to patients and all participated in our study. However, only 6 nurses provided responses, which could be used for the data analysis, 3 nurses provided little or no answers to the open-ended questions, which form the basis of this article. In line with the recommendation for qualitative research (source: https://doi.org/10.1177%2F0034355213502549, https://doi.org/10.1093/intqhc/mzm042 ), we have decided to only report results for the datasets which contained enough “data richness” for qualitative analysis.
You asked about the availability of the education materials, that several nurses mentioned: the materials were designed as part of the iCARE-PD project and are therefore subject to copyright regulations, which is why it was not possible for us to publish the materials. However, the consortium plans to make these materials freely available at the end of the study.
Based on the other reviews, we have revised our limitation section and formulated our results with more caution. Also, we have revised the text and improved its readability.
We hope that we have addressed your comments sufficiently.
Kind regards,
Marlena van Munster
Reviewer 2 Report
Telemedicine, according to mainstream approach, is when a doctor prescribes a specific remote follow-up with bidirectional audio video streaming protocol using a registered "medical device" of at least a "medical software" in which also nursing may play a relevant role. Curricula for telemedicine shall be provided during the medical school.
As far as I understood the study consisted of 6 opinion leaders that were interviewed regarding PN approach to telemedicine according to a survey. Only a part of the experts replied the survey.
The experts in PD nursing declared from 1 to 3 years of experience in the field. This seems someone at the very beginning of their career.
This seems a very preliminary inquiry in the field. The overall interest of the work on Parkinson seems marginal, not a single disease specific concept is analyzed (telemedicine and analysis of barriers in the environment of the patients, drug monitoring, syndrome fluctuation, caregiver training) and moreover surprisingly authors' stated how "The role of the PN has not yet been clearly defined"
The limits of the survey must be better explained to the readers, also the mean age and nationality of the experts may be interesting to understand how the concepts explained are generalizable.
Author Response
Dear Expert,
Thank you for dedicating your time and expertise to reviewing our manuscript.
We agree that the sample size is rather small, however, we still think that the provision of practice insights is adding a valuable contribution to the field. In the care model, we have interviewed 9 Parkinson Nurses (not opinion leaders) providing telemedicine care to patients and all participated in our study. However, only 6 nurses provided responses, which could be used for the data analysis, 3 nurses provided little or no answers to at least one open-ended question, which form the basis of this article. In line with the recommendation for qualitative research (source: https://doi.org/10.1177%2F0034355213502549, https://doi.org/10.1093/intqhc/mzm042 ), we have decided to only report results for the datasets which contained enough “data richness” for qualitative analysis.
We agree that a sample description helps to better understand the results, which is why we have added information about the sample in Table 2.
Based on your and the other reviews, we have revised our limitation section and formulated our results with more caution. Also, we have revised the text and improved its readability.
We agree that the role of PN is well defined in the UK and Northern America, however, for many parts of Europe, this is not the case (for reference see: https://doi.org/10.1024/1012-5302/a000617, https://doi.org/10.3390/geriatrics7020046). PN are not regularly present as part of the care team and in many countries and a multidisciplinary approach to Parkinson´s care is not the standard practice. For PNs, there is a lack of employment/ training opportunities and clear competency descriptions, which lead us to our statement, and is why we think that more research on this topic is needed.
We hope that we have addressed your comments sufficiently.
Kind regards,
Marlena van Munster